# Spatial Scale Effect of a Typical Polarized Remote Sensor on Detecting Ground Objects

**DOI:** 10.3390/s21134418

**Published:** 2021-06-28

**Authors:** Ying Zhang, Jingyi Sun, Rudong Qiu, Huilan Liu, Xi Zhang, Jiabin Xuan

**Affiliations:** School of Instrumentation Science & Opto-Electronics Engineering, Beihang University, No. 37 Xueyuan Road, Haidian District, Beijing 100191, China; yingzhang@buaa.edu.cn (Y.Z.); jingyisun@buaa.edu.cn (J.S.); liuhuilan@buaa.edu.cn (H.L.); zhangxi@buaa.edu.cn (X.Z.); xuanjiabinbj@163.com (J.X.)

**Keywords:** spatial scale effect, polarized remote sensor, spatial heterogeneity, super resolution image reconstruction

## Abstract

For polarized remote sensors, the polarization images of ground objects acquired at different spatial scales will be different due to the spatial heterogeneity of the ground object targets and the limitation of imaging resolution. In this paper, the quantitative inversion problem of a typical polarized remote sensor at different spatial scales was studied. Firstly, the surface roughness of coatings was inversed based on the polarized bidirectional reflectance distribution function (pBRDF) model according to their polarization images at different distances. A linear-mixed pixel model was used to make a preliminary correction of the spatial scale effect. Secondly, the super-resolution image reconstruction of the polarization imager was realized based on the projection onto convex sets (POCS) method. Then, images with different resolutions at a fixed distance were obtained by utilizing this super-resolution image reconstruction method and the optimal spatial scale under the scene can be acquired by using information entropy as an evaluation indicator. Finally, the experimental results showed that the roughness inversion of coatings has the highest accuracy in the optimal spatial scale. It has been proved that our proposed method can provide a reliable way to reduce the spatial effect of the polarized remote sensor and to improve the inversion accuracy.

## 1. Introduction

Polarization, as one of the characteristics of light, can carry additional information about objects it acts on. This makes it widely used in many kinds of imaging and non-imaging remote sensing research to detect the information that is difficult to be detected by traditional methods [1]. With the expansion of the demand for remote sensing information, polarized remote sensing information can solve problems that cannot be solved in the traditional remote sensing field, and effectively improve the accuracy of feature recognition. Thus, it has become a valuable tool in many applications, including cloud and atmospheric aerosol detection, geological exploration, soil analyses, and medical diagnoses [2,3,4,5].

With the widespread use and maturity of polarized remote sensing, whether near-ground or remote sensing platform polarization detection, polarization imaging quantitative remote sensing has become one of the main trends in the development of remote sensing. However, when using the polarized remote sensing data acquired by polarized sensors, the characteristic parameters of ground objects must be inverted quantitatively. The inversion accuracy will be affected by several factors, especially the spatial scale effect, meaning the measurement results will be changed according to the selected measurement scale. Generally, spatial scale refers to the spatial resolution of an image. At different spatial scales, the inversion or classification accuracy varies significantly. This affects judgements concerning the impact of the classification of terrain, spring phenology detections, the estimation of the fire range, and the inversion of vegetation evaporation [4,5,6,7,8,9]. Additionally, the phenomena and summary rules on a specific scale may be valid on another scale, but it is not entirely applicable. Therefore, addressing the impact of this effect is particularly important for improving the inversion accuracy based the detection of polarized sensors.

Currently, there are two main methods available for reducing this effect: correcting the scale effect and detecting in the optimal spatial resolution. The spatial heterogeneity on sub-pixels is parameterized and the expression of normalized difference vegetation index (NDVI) is corrected by covariance and variance, so that it does not change with scale [10]. Through the new semi-empirical parameters obtained by the least squares method, the leaf area index (LAI) and the effective photosynthetic radiation absorbed by plants suitable for small-scale remote sensing were extended to be globally applicable [11]. The parameters of the structural features were calculated by the area ratio of various types of features and utilized to correct the error caused by the scale effect in LAI [12,13]. In terms of scale correction, the proposed method could be applied to specific parameters, but whether the method was applicable to other parameters was not mentioned. To solve this problem, some scholars proposed detection at suitable or optimal scales. In 2004, South Korean scientists Saro Lee et al. [14] studied the influence of spatial resolutions on landslide threat determination by re-sampling aerial remote sensing data, and found that an image with a spatial resolution of at least 30 m was needed to determine the landslide. In 2011, Ming [15] proposed an improved local variance method for the optimal scale selection of remote sensing images and found that the statistical results are more obvious and can obtain a more reasonable optimal scale range. In 2013, Angela Lausch et al. [16] studied the effects of different spatial resolutions on the NDVI to obtain a suitable scale. In 2019, Guo et al. [17] investigated the scale effects by analyzing the linear relationships between VI calculated from red-green-blue (RGB) images from unmanned aerial vehicles (UAV) and ground leaf chlorophyll contents of maize measured using SPAD-502.

To obtain multiscale images, upscaling is necessary. In this context, we apply super-resolution image reconstruction in upscaling, to achieve a low to high resolution extension. The theoretical basis of the single-frame super-resolution reconstruction technique is to establish the corresponding low and high-resolution image set; to calculate the correspondence between them to obtain prior knowledge; and to restore the lacking high frequency information using the sparse representation method [18,19,20]. However, it is difficult to acquire the low- and high-resolution image set, simultaneously. Thus, multi-frame super-resolution image reconstruction technology was utilized in the study. This process relies on multiple low spatial resolution sequence images, which are utilized to acquire a higher resolution image beyond that of the original optical imaging system, through image interpolation, image registration, and image reconstruction. The idea was first proposed by Tsai and Huang [21] and was successfully applied in remote sensing images using the frequency domain method. Spatial heterogeneity, the difference of the internal composition, or the brightness distribution of the pixel, are also ubiquitous.

Therefore, we put forward our proposal for reducing the spatial scale effect, by considering the combination of spatial heterogeneity of ground object targets and the limitation of imaging resolution. In this paper, the quantitative inversion problem of a typical polarized remote sensor at different spatial scales was studied by taking into account the surface roughness of the coating. The remote sensor is based on Liquid Crystal Variable Retarder (LCVR), which can achieve a high polarization measurement accuracy. The design and calibration of this typical polarized remote sensor have been studied before [22]. Here, the spatial scale effect of the sensor on the roughness inversion of the coating was analyzed from the perspective of spatial heterogeneity and our spatial scale effect correction method for linear mixed pixels was proposed. Moreover, the optimal spatial scale was found based on super-resolution image reconstruction using entropy as an evaluation criterion. Research on super-resolution polarization image reconstruction technology is of great significance to the application of polarization imaging quantification inversion based on the detection of typical polarized sensors.

This paper is organized into 4 sections. Following the introduction in Section 1, Section 2 describes the methods to invert the target characteristic, reduce the spatial scale effect for linear mixed pixels, and reconstruct super-resolution images for the optimal spatial scale. The corresponding experimental preparation and results are presented in Section 3. Finally, Section 4 concludes the paper.

## 2. Methods

### 2.1. Inversion Model Based on pBRDF Model

To complete the inversion of the coating roughness, we must implement two steps: the establishment of the pBRDF coating model and the acquisition of the target polarization state, through the polarization images.

#### 2.1.1. Modeling of pBRDF

In order to study the physical properties of an object, for a given incident state, the pBRDF models of the target are necessary to predict the polarization state of the reflected light, which is a prerequisite for the quantitative analysis of polarimetric images in remote sensing.

In this paper, the pBRDF model, based on Priest and Gerner’s [23] microfacet theory, is utilized, but the complicated description of diffuse reflection has been simplified. This theory assumes that a rough surface, with roughness larger than or equal to the detection wavelength, is composed of a collection of randomly oriented microfacets [24]. Each microfacet acts as a specular reflector obeying Fresnel’s equations, and has a certain rule of normal distribution, which is generally considered to be Gaussian distribution [25]:(1)$$p\left(\theta \right)=\frac{1}{2\pi \sigma^{2}\cos^{3}(\theta )}\exp(\frac{-\tan^{2}(\theta )}{2\sigma^{2}})$$
where *σ* is the root mean square slope [26], and *θ* is the orientation angle of microfacets, relative to the object surface normal. *θ* can be defined as:(2)$$\cos(\theta )=\frac{\cos(\theta_{i})+\cos(\theta_{r})}{2\cos(\beta )}$$
(3)$$\cos(2\beta )=\cos(\theta_{i})\cos(\theta_{r})+\sin(\theta_{i})\cdot \sin(\theta_{r})\cos(\varphi_{i}-\varphi_{r})$$
where the incident zenith and azimuth angles are given by *θ_i_* and *φ**_i_,* respectively, and the reflected zenith and azimuth angles are given by *θ_r_* and *φ_r_*, respectively. The angle relationship between the microelement surface and the object surface is shown in Figure 1 [27].

According to the physical meaning of bidirectional reflectance distribution function (BRDF), it is the ratio of the radiance L, reflected on the object surface, and the irradiance E, incident on the object surface, in a particular direction. The geometry of the BRDF definition is illustrated in Figure 2.

Extended to the field of polarization, polarized BRDF, also known as pBRDF, means that in the physical sense of BRDF, both radiance and irradiance are represented by a 4 × 1 Stokes vector, and F is a 4 × 4 Mueller matrix. Since a Mueller matrix relates the incident and reflected Stokes vectors, the polarimetric relationship between the incident and reflected light can be given by:(4)$$dL_{r}\left(\theta_{r},\varphi_{r}\right)=F\left(\theta_{i},\varphi_{i},\theta_{r},\varphi_{r},\lambda \right)dE\left(\theta_{i},\varphi_{i}\right)$$
where *L_r_* and *E* are the reflected and incident Stokes vector, respectively.

For the coating, the reflection consists mainly of mirror and diffuse reflection. It is generally believed that specular reflection carries polarization information, due to the single scattering event on the coated surface. Meanwhile the diffuse reflection part becomes Lambertian reflection after multiple scattering event, on or inside the coated surface, which has no polarization characteristics and is independent of reflection angle. Therefore, the pBRDF Mueller matrix can be expressed as [28]:(5)$$F\left(\theta_{i},\theta_{r},\varphi_{i}-\varphi_{r},\sigma ,\varepsilon ,k_{d}\right)=\frac{1}{2\pi }\frac{1}{4\sigma^{2}}\frac{1}{\cos^{4}(\theta )}\;\frac{\exp(-\frac{\tan^{2}(\theta )}{2\sigma^{2}})}{\cos(\theta_{r})\cos(\theta_{i})}M\left(\theta_{i},\theta_{r},\varphi_{i}-\varphi_{r},\varepsilon \right)G\left(\theta_{i},\theta_{r}\right)+\frac{k_{d}}{\cos(\theta_{i})}\varphi$$
where *G* is the shadowing/masking factor [29,30], *M* is the Mueller matrix for the Fresnel reflection from each facet with the same size as *F*, *ε* is the complex refractive index of the target, and *k_d_* is the diffuse reflection coefficient.

In this model, three target parameters are involved, including the real and imaginary parts of the index of refraction *n* and *κ*, respectively, and the surface roughness *σ*. The pBRDF Mueller matrix also has properties of anisotropy related to the observed geometry.

#### 2.1.2. Acquisition of the Target pBRDF

Through the pBRDF model, the theoretical polarization state of reflected light (natural light reflected from the coating) can be obtained. Here, we explain how to acquire the polarization state of the reflected light at the entrance pupil, through the polarization image captured by the owned polarization imaging detector [31]. The detector consists of a filter, two LCVRs, and a polarizer.

In this experiment, the light source is unpolarized, so its Stokes vector is assumed to be (1 0 0 0) ^T^. Also, the coating acts as a polarizer, which makes the reflected light carry the target information. *S_in_* is the Stokes vector of light reflected by the coating, while *S_out_* is the Stokes vector of the emitted light passing through the filter, LCVR and polarizer. The relationship between the two can be described as:(6)$$S_{out}=M_{p2}\times M_{LCVR2}\times M_{LCVR1}\times M_{p1}\times S_{in}$$
where *M*_*p*1_ and *M*_*p*2_ represent the Mueller matrix of the filter and the polarizer, respectively, and *M*_*LCVR*1_ and *M*_*LCVR*2_ is the Mueller matrix of two LCVRs. Combining into the Mueller matrices of the instrument M_ins_, it can be expressed as:(7)$$S_{out}=M_{ins}\cdot S_{in}$$

Therefore, the Stokes vector of the light reflected from the coating can be expressed as:(8)$$S_{in}=M_{ins}{}^{-1}\cdot S_{out}$$

M_ins_ is calibrated by the method based on both a multi-band light source and a laser source for liquid crystal variable retarders in advance [32]. Meanwhile, the Stokes vector of the light reflected from the coating can be also expressed as:(9)$$S_{in}=\vec{f_{r}}\cdot {\left(\begin{array}{cccc} 1 & 0 & 0 & 0 \end{array}\right)}^{T}$$
where $\vec{f_{r}}$ is a 4 × 4 Mueller matrix of the coating, that is, the pBRDF matrix. However, since the Stokes vector of unpolarized light source is (1 0 0 0) ^T^, only the first column of $\vec{f_{r}}$ needs to be considered, namely, ${\left(\begin{array}{cccc} f_{00} & f_{10} & f_{20} & f_{30} \end{array}\right)}^{T}$.

Using the diffuse reflectance standard (DFS) as a calibration target, the pBRDF of any painted surface can be calculated with the comparison method. The DFS has a highly Lambertian reference surface, and an approximately angular-invariant BRDF of *ρ/π* with a nearly randomly polarized reflectance of *ρ*. Thus, the pBRDF of the coating can be expressed as:(10)$$\left[\begin{array}{c} f_{00}\\ f_{10}\\ f_{20}\\ f_{30} \end{array}\right]=\frac{\rho }{\pi }\left[\frac{B\prime }{A\prime }\right]$$
where A′ and B′ are the Stokes vector of the DFS and target calculated by Equation (8), respectively.

Relevant scientific research shows the $f_{30}$ in pBRDF is generally not considered because its value is generally small and negligible. Moreover, $f_{10}$ and $f_{20}$ are easily affected by uneven coating spray. Thus, $f_{00}$ is used as an inversion reference in the subsequent roughness inversion of the coating.

#### 2.1.3. Levenberg–Marquardt Algorithm for Inversion

In the microfacet pBRDF model, under the same incident and observation conditions, there are and only three unknown parameters, the real n and imaginary k parts of the complex refractive index, and the surface root mean square slope σ. In order to realize parameter inversion, the objective function is inverted through a Levenberg–Marquardt (LM) algorithm as follows:(11)$$\min f\left(n,k,\sigma \right)=\frac{\sum_{\theta_{i}}\sum_{\theta_{r}}\sum_{\varphi }{\left[f_{s}^{m}\left(\theta_{i},\theta_{r},\varphi \right)-f_{s}\left(\theta_{i},\theta_{r},\varphi ,n,k,\sigma \right)\right]}^{2}}{\sum_{\theta_{i}}\sum_{\theta_{r}}\sum_{\varphi }{\left[f_{s}^{m}\left(\theta_{i},\theta_{r},\varphi \right)\right]}^{2}}$$
where $f_{s}\left(\theta_{i},\theta_{r},\varphi ,n,k,\sigma \right)$ is the simulation value with Equation (5), and $f_{s}^{m}\left(\theta_{i},\theta_{r},\varphi \right)$ is calculated by Equation (8).

### 2.2. Spatial Scale Effect Correction for Linear Mixed Pixels

In a pixel of the same paint coating, multiple sub-pixels with different roughness appear. However, the surface characteristics can be described by the pBRDF model proposed above, due to the definition of linear mixed pixels.

The quantitative relationship between pBRDF values and roughness σ can be expressed as:(12)$$f_{r}=f\left(\sigma \right)$$

Inverting the function of Equation (12), we get:(13)$$\sigma_{r}=g\left(f_{r}\right)$$

Given that the six input parameters $\left(\theta_{i},\;\theta_{r},\;\varphi_{i},\;\varphi_{r},\;\varepsilon ,\;k_{d}\right)$ are known, Equation (13) indicates that we can calculate the corresponding pixel roughness value from the measured pBRDF value in a certain pixel. The roughness value only represents the average roughness in a certain area.

When there is only one type of sub-pixel target in the pixel, that is, the main sub-pixel target, the average roughness can indicate the roughness of the main target. However, it is possible a pixel contains one kind of major sub-pixel target with multiple interfering sub-pixel targets, as shown in Figure 3.

Thus, the value $f_{r}$ measured by the pixel can be expressed as a weighted sum of the values of the respective interfering sub-pixels and the main part of the pixel by the weight of the area, such that:(14)$$f_{r}=\overset{n}{\underset{i=1}{\sum }}a_{i}f\left(\sigma_{i}\right)=\overset{n}{\underset{i=1}{\sum }}a_{i}f_{i},i=1,2,\ldots ,n\;\overset{n}{\underset{i=1}{\sum }}a_{i}=1,i=1,2,\ldots ,n$$
where $f_{i}$ is the pBRDF value, $\sigma_{i}$ is the roughness and $a_{i}$ is the proportion of the area occupied by each of the corresponding sub-pixel targets *i*.

Therefore, the actual measured roughness value $\sigma_{a}$ can be expressed as:(15)$$\sigma_{a}=\overset{n}{\underset{i=1}{\sum }}a_{i}g\left(f_{i}\right),i=1,2,\ldots ,n$$

Generally, $\sigma_{r}$ in Equation (13) and $\sigma_{a}$ in Equation (15) are not equal, due to the scale effect. When studying the error of $\sigma_{r}$ and $\sigma_{a}$, we can only take the quadratic approximation of the Taylor expansion of the Equation (15), such that:(16)$$\sigma_{a}=\sigma_{r}+\sigma_{r}^{\prime }\overset{n}{\underset{i=1}{\sum }}a_{i}\left(f_{i}-f_{r}\right)+\frac{1}{2}\sigma_{r}^{\prime \prime }\overset{n}{\underset{i=1}{\sum }}a_{i}{\left(f_{i}-f_{r}\right)}^{2},i=1,2,\ldots ,n$$

Note that this is the second-order approximation. The second item in Equation (16) can be expanded:(17)$$\overset{n}{\underset{i=1}{\sum }}a_{i}\left(f_{i}-f_{r}\right)=\overset{n}{\underset{i=1}{\sum }}a_{i}f_{i}-\overset{n}{\underset{i=1}{\sum }}a_{i}f_{r}=f_{r}-f_{r}=0$$

The second term in Equation (16) is zero. The third term in Equation (16) represents the local variance, which is different from variance in the traditional sense, and can be specifically defined as:(18)$$D\left(f\right)=\overset{n}{\underset{i=1}{\sum }}a_{i}{\left(f_{i}-f_{r}\right)}^{2}$$
where D(f) is the local variance of the roughness distribution of mixed pixels, which can be described as the square error between the central pixel value and the mean value of surrounding pixels. The physical meaning is the difference between the detection values of the central and surrounding pixels, that is, the difference in the sub-pixel composition.

Substituting Equations (17) and (18) into Equation (16), the scale effect correction formula for the surface roughness of linear mixed pixels can be obtained by:(19)$$\sigma_{a}=\sigma_{r}+\frac{1}{2}\sigma_{r}^{\prime \prime }D\left(f\right)$$

In Equation (19), $\sigma_{r}^{\prime \prime }D\left(f\right)$ denotes the scale effect error produced by the linear mixed pixel, including the $\sigma_{r}$ parameter related to the nonlinearity of the model, and the D(f) parameter related to spatial heterogeneity. The domain range of the target pixel can be selected based on the size of the target, to calculate the value of D(f). A square with a side length of 3 or 5 pixels, centered on the target pixel, is advisable.

### 2.3. Super-Resolution Image Reconstruction Based on the POCS Method

Super-resolution image reconstruction meets the needs of multi-resolution, to find the optimal scale, and further improves the comprehensive quality of images, compared to traditional image resampling [33].

The general model for generating low-resolution images, from ideal or high-resolution images, is shown in Figure 4. It describes the geometric transformation of continuous high-resolution scenes, from world coordinates to camera coordinates; the loss of spatial resolution, due to blurring caused by the camera’s point spread function; and spatio-temporal sampling, while introducing some noise into the process. Its inverse process is the super-resolution image reconstruction. In this paper, the super-resolution image reconstruction technology used is mainly based on the POCS algorithm [34,35].

In the super-resolution image reconstruction process, an image acquisition model needs to be constructed, to connect the original high-resolution and observed low-resolution image. A general model can be expressed as:(20)$$g_{l}\left(m_{1},m_{2}\right)=\underset{\left(n_{1},n_{2}\right)}{\sum }f\left(n_{1},n_{2}\right)h_{l}\left(m_{1},m_{2};n_{1},n_{2}\right)+\eta_{l}\left(m_{1},m_{2}\right)$$
where $g_{l}\left(m1,m2\right)$ is the first frame of the observed low-resolution image, f(n1,n2) is the initial high-resolution image, $h_{l}\left(m1,m2;n1,n2\right)$ is the spatial point spread function, and $\eta_{l}\left(m1,m2\right)$ is the additive noise.

In the POCS method, the solution of POCS is limited to a closed convex set *C_i_*, by each constraint or prior knowledge. For each *C_i_*, there is a corresponding projection operator *P_i_.* A non-zero solution space is defined as:(21)$$f\in C_{0}=\overset{i=m}{\underset{i=1}{\cap }}C_{i}P_{i},i=1,2,\ldots ,m$$
where *m* is the number of closed convex sets, corresponding to *m* prior knowledge. The process of iteratively solving the original image according to the convex set constraint is as follows:(22)$$f_{k+1}=T_{m}T_{m-1}\ldots T_{1}f_{k},k=1,2,\ldots$$
(23)$$T_{i}=\left(1-\lambda_{i}\right)I+\lambda_{i}P_{i},0<\lambda_{i}<2$$
where *T_i_* is the relaxation projection operator corresponding to the prior condition and *λ_i_* is constant. The convergence speed of the algorithm can be adjusted by changing the value of *λ_i_*.

Assuming the noise follows a Gaussian distribution, the variance $\sigma_{r}$ and a reasonable statistical confidence coefficient *c* is given, from which the a priori boundary, $c\sigma_{r}(c\geq 0)$, can be obtained. Each element in the image $M_{1}\cdot M_{2}$ needs to satisfy the following conditions:(24)$$C_{m_{1},m_{2}}=\left\{f\left(n_{1},n_{2}\right):r^{\left(y\right)}\left(m_{1},m_{2}\right)\leq c\sigma_{r}\right\}\;0\leq m_{1}\leq M_{1}-1\;0\leq m_{2}\leq M_{2}-1$$
where $c\sigma_{r}$ is the a priori boundary, which statistically reflects that $f\left(n_{1},n_{2}\right)$ is an element in $C_{m1,m2}$
(25)$$r^{\left(y\right)}\left(m_{1},m_{2}\right)=g_{l}(m_{1},m_{2})-\sum_{n_{1}=0}^{M_{1}-1}\sum_{n_{2}=0}^{M_{2}-1}f\left(n_{1},n_{2}\right)h\left(m_{1},m_{2},n_{1},n_{2}\right)$$
where $r^{\left(y\right)}\left(m_{1},m_{2}\right)$ is the residual between $f\left(n_{1},n_{2}\right)$ and $g_{l}\left(m_{1},m_{2}\right)$.

By comparing Equations (25) and (20), we can conclude that the residual should be consistent with the noise. That is to say, the statistical process of noise can be utilized to define the difference between the original and current image. In each iteration, the absolute value of the residual is also within the bounds, like the noise.

The expression that projects $f\left(n_{1},n_{2}\right)$ to $C_{m_{1},m_{2}}$ can be defined as:(26)$$y\left(n_{1},n_{2}\right)=P_{m_{1},m_{2}}\left[f\left(n_{1},n_{2}\right)\right]=f\left(n_{1},n_{2}\right)+\;\left\{\begin{array}{cc} \frac{\left[r^{\left(y\right)}\left(m_{1},m_{2}\right)-\delta_{0}\right]}{\sum_{x_{1}}\sum_{y_{1}}h_{k}^{2}\left(m_{1},m_{2};x_{1},y_{1}\right)}h_{k}\left(m_{1},m_{2};n_{1},n_{2}\right) & r^{\left(y\right)}(m_{1},m_{2})>\delta_{0}\\ 0 & \left|r^{\left(y\right)}\left(m_{1},m_{2}\right)\right|\leq \delta_{0}\\ \frac{\left[r^{\left(y\right)}\left(m_{1},m_{2}\right)+\delta_{0}\right]}{\sum_{x_{1}}\sum_{y_{1}}h_{k}^{2}\left(m_{1},m_{2};x_{1},y_{1}\right)}h_{k}\left(m_{1},m_{2};n_{1},n_{2}\right) & r^{\left(y\right)}\left(m_{1},m_{2}\right)<-\delta_{0} \end{array}\right.$$

According to the above principle, the implementation idea of the POCS algorithm can be divided into three steps. Firstly, a reference frame image is selected from low-resolution image sequences and is converted into an initial estimate of the high-resolution image, by some interpolation method. Image registration is then performed on other low-resolution images, to obtain corresponding motion parameters. Finally, the point spread function (PSF) is used for iteration to correct the initial estimate of the high-resolution image, until an acceptable reconstruction result is obtained that satisfies the threshold. The whole process is illustrated in Figure 5.

## 3. Experimental Validation and Results

### 3.1. Experiment 1: Spatial Scale Effect Correction

#### 3.1.1. Experimental Preparation and Scheme

To verify the validity of our proposed spatial scale effect correction method, we carried out an indoor experiment. We chose two targets with different surface roughness. These are shown in Figure 6. The target is a standard compact disc (CD). The bottom surface was unchanged, while the top surface had matte white paint, of different thickness, sprayed onto it. The different thickness created a coating surface with different roughness. The corresponding sampling length was selected according to ISO 4288-1985. The average value of multiple measurements was taken as the target roughness using the roughness tester.

The polarimetric images of targets were captured using our LCVR-based polarized remote sensor. Moreover, all images were taken in 514 nm. A standard source integrating sphere (Labsphere US-120-SF) was used in conjunction with the light source. It produced a source stability better than 0.001 and could be considered as a uniform non-polarized incident light.

The experimental scheme is shown in Figure 7. The target position was stationary, while the position of the image polarizer was adjusted to receive the reflected light along the 45° reflection angle of the coating surface. Meanwhile, the heights of the image polarizer, target and source were consistent with the ground. This was to ensure the relative azimuth of all pixels was greater than 170°. Multiple polarization images of coatings with uneven roughness were captured at five distances. At each distance, only the incident angle of the light source was changed, and 13 datasets were measured from 39° to 51°, with 1° intervals, for inverting the roughness of the coating surface. To ensure the accuracy of the change in incident light angle, the rotation angle of the light source was adjusted by the laboratory electric turntable, with a precision of 0.1°. Finally, the surface roughness of the coatings was inverted based on the pBRDF model and corrected by our proposed method.

#### 3.1.2. Results and Analysis

The premise of our proposed method in this paper is that the interfered pixels account for a small section of all pixels. Therefore, the mean and variance of the degree of polarization (DOP) data, of the target coating, can be used to reflect the correction effect. An area where the partial roughness is uneven results in the maldistribution of the DOP, which increases the overall variance of the DOP. The method in this paper only corrects the DOP value of the partially uneven area. Therefore, the overall DOP average of the image before and after the correction should be approximately equal. The variance of the DOP indicates the spatial heterogeneity. The main function of the scale effect correction, proposed in this paper, is to reduce the influence of spatial heterogeneity. Thus, the variance of DOP after correction should be significantly reduced.

Taking the data measured at an incident angle of 45° as an example, the mean and variance of the DOP, before and after the correction at each distance, are calculated and listed in Table 1. It can be seen that after the scale effect correction, the DOP mean value does not change by more than 0.2%, while the DOP variance decreases by an average of 13.5%. This shows that after the correction, the DOP gap between the pixels, at each scale, has been significantly reduced, indicating that the impact of spatial heterogeneity has decreased. Furthermore, the inversion results of the different distances between the imager and the targets are shown in Table 2, to further verify the correction effect.

In each group, the corrected roughness value(σ_a_) is no greater than the uncorrected value (σ_r_) directly inverted from the pBRDF model, and closer to the real roughness. Combined with the change of DOP variance in Table 1, we conclude that our method can reduce the influence of spatial heterogeneity on the scale effect and improve the accuracy of the inversion. Moreover, our method takes the average roughness of all pixels within the detection area as the main target. Therefore, it may be affected by extreme values, resulting in some overcorrection phenomenon.

Since our camera is a prime, distance is proportional to the spatial resolution. From the corrected data, the influence of spatial resolution on the inversion accuracy can also be found in Table. The inversion accuracy is the highest only at the optimal resolution. Therefore, we propose a new method to find the optimal resolution, or the optimal scale.

### 3.2. Experiment 2: Super-resolution Reconstruction at the Optimal Scale

#### 3.2.1. Experimental Preparation and Scheme

We used the existing instrument (described in Section 3.1.1) to capture multi-frame polarized images in 514 nm at the distances of 7.5 m, 10 m and 15 m. To minimize the environmental impact, we chose to carry out the experiment at night, and with a larger target 3, shown in Figure 8. The same paint was evenly sprayed at the center of the target, consisting of the numbers 1–8. The target was 1 m × 1 m in size. The light source was a tungsten halogen lamp, which has no polarization property and is approximately considered to be natural light.

The experimental scheme is shown in Figure 9. A multi-frame image of the same scene at different detection distances was taken, and then a DOP image of each frame was calculated. Next, these polarization images were used as input frames for super-resolution image reconstruction, to obtain high-resolution polarized images of different magnification factors. The strict relationship between magnification and the actual spatial scale was not requisite. It was sufficient knowing they are inversely proportional, and it is feasible to obtain different spatial scales by reconstructing the images with different magnification factors. During super-resolution image reconstruction, we used 24 frames of the low-resolution image to obtain the corresponding high-resolution image. Meanwhile, multi-scale images of the DFS were obtained in preparation for inversion by image interpolation. Image entropy was used as the evaluation criterion, for preliminarily selecting the optimal scale. Finally, the target coating roughness of the multiple areas enclosed by red boxes in Figure 8, were inverted using Equation (11). After this, the optimal scale obtained, in the previous step, was verified.

#### 3.2.2. Result and Analysis

Using super-resolution image reconstruction method based on POCS, DOP images of different scales were generated. The resolution of the newly generated image was 1–5 times that of the original image. As an example, the multiscale DOP images taken at the distance of 10 m were reconstructed and are shown in Figure 10. It can be seen that the reconstructed images have a higher contrast than the original image and are richer in detail.

To select the optimal scale, the statistical entropy calculated by the grayscale co-occurrence matrix is regarded as an index to evaluate the amount of information of the reconstructed DOP image texture [36]. The statistical entropy can be expressed as:(27)$$ENT=-\overset{L-1}{\underset{i=0}{\sum }}\overset{L-1}{\underset{j=0}{\sum }}P\left(i,j\right)\mathrm{lg}P\left(i,j\right)$$
where *P(i,j)* represents the probability of occurrence of grayscale pairs in the image, and L represents the total number of gray levels.

The score for different magnification factors, at multiple distances, is shown in Figure 11a. As the magnification factor increases, the entropy value reaches its peak, indicating that the spatial correlation between the pixels is weakest at this spatial scale. These may be the optimal scales at the selected distance.

The interval where the best value is located can be acquired, but the exact optimal scale could not be obtained due to the coarseness of our chosen magnification factors. At the distances of 7.5 m, 10 m and 15 m, the optimal scale was located in the magnification factor ranges of 3–5, 2–4 and 1–3, respectively. As the magnification factor increases to 5, the value for the shorter distances remains at a higher level, while the farthest becomes lower, than the original value without reconstruction. There are two possible reasons for this. One is that the images could be reconstructed with a limited multiple by the selected POCS algorithm. The other is that the optimal scale is inversely proportional to the distance. Furthermore, the optimal scale is the only preliminary result that must be verified by the inversion results.

In Figure 11b, the variation of the difference between the inversion and real values, with amplification factor at the same distance, is almost in alignment with that of the image entropy value shown in Figure 11a. The peak locations are also similar, in terms of the magnification factor ranges. The inverted and measured values of σ are detailed in Table 3. At the appropriate scale, the inversion accuracy of σ has improved significantly. The optimal magnification factor for 7.5 m, 10 m and 15 m is 4, 3 and 3, respectively, and the value of the inversion error is 6.45%, 0.95% and 3.95%, respectively. This is consistent with the preliminary result. Furthermore, comparing to before reconstruction, the error reduced by 27.66%, 9.8% and 28.77% at the optimal scales, respectively. Thus, our method is capable of finding the optimal scale at a fixed distance and can greatly improve the inversion accuracy.

## 4. Discussion

Two methods to reduce the spatial scale effect of the typical polarized remote sensor were proposed for two different types of target coatings. For the coatings lacking obvious features, such as targets 1 and 2, the spatial scale effect caused by the difference in sub-pixel distribution was analyzed. We implemented a scale correction formula based on the local variance of the roughness, to modify the inverted value obtained by the modified pBRDF model. Moreover, the inversion accuracy at each scale was improved before correction. However, for the target 3, which has distinct features and was regarded as containing nonlinear mixed pixels, this correction formula was not appropriate. Instead, we used a new method to find the optimal scale through super-resolution reconstruction, based on the POCS method. Multiscale high-resolution polarization images, obtained from a series of low-resolution images, were prepared as the initial images of the inversion. Using the image entropy value as the evaluation criterion, the preliminary results of the optimal scale were obtained and verified. We provide some suggestions for choosing the right spatial resolution for different application goals and requirements. A follow-up study will include the acquisition of the exact optimal scale, and super-resolution image reconstruction on images without obvious features.

## 5. Conclusions

In this paper, accessible methods to reduce the spatial scale effect of a typical polarized remote sensor were proposed. For the coatings lacking obvious features, a scale correction formula based on the local variance of the roughness was implemented, to modify the inverted value obtained by the modified pBRDF model. For the targets with distinct features, a super-resolution polarization image reconstruction algorithm based on POCS was proposed to obtain higher resolution polarization images, and related research on the selection of optimal scale for quantitative inversion of coating roughness was carried out. Moreover, the image quality of DOP images in different scales was evaluated, which verified the effectiveness of the reconstruction algorithm used in this paper for scaling transformation. Further research will seek to improve the image reconstruction algorithm.

## Figures and Tables

**Figure 1 sensors-21-04418-f001:**
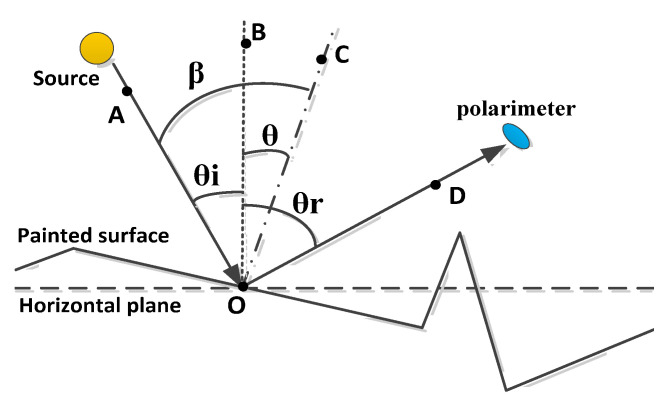
Angle definitions in a microfacet coordinate system; where OB is the surface normal, OC is the target microfacet normal, AO is the incident direction, and OD is the detection direction [27].

**Figure 2 sensors-21-04418-f002:**
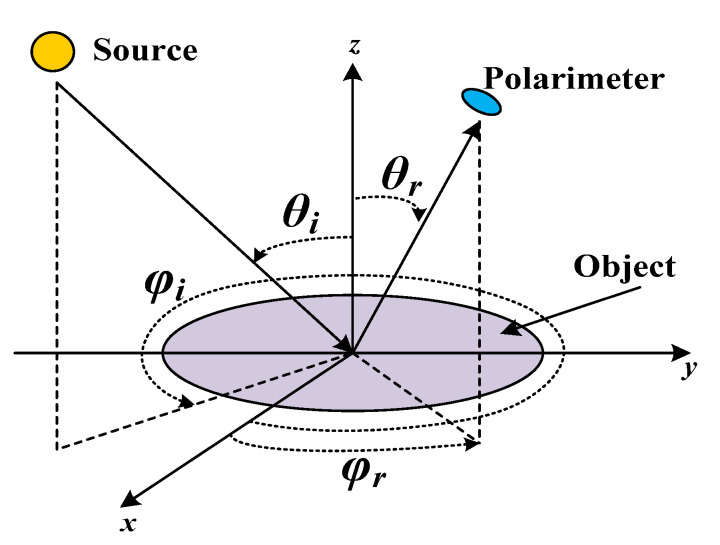
BRDF geometry [27].

**Figure 3 sensors-21-04418-f003:**
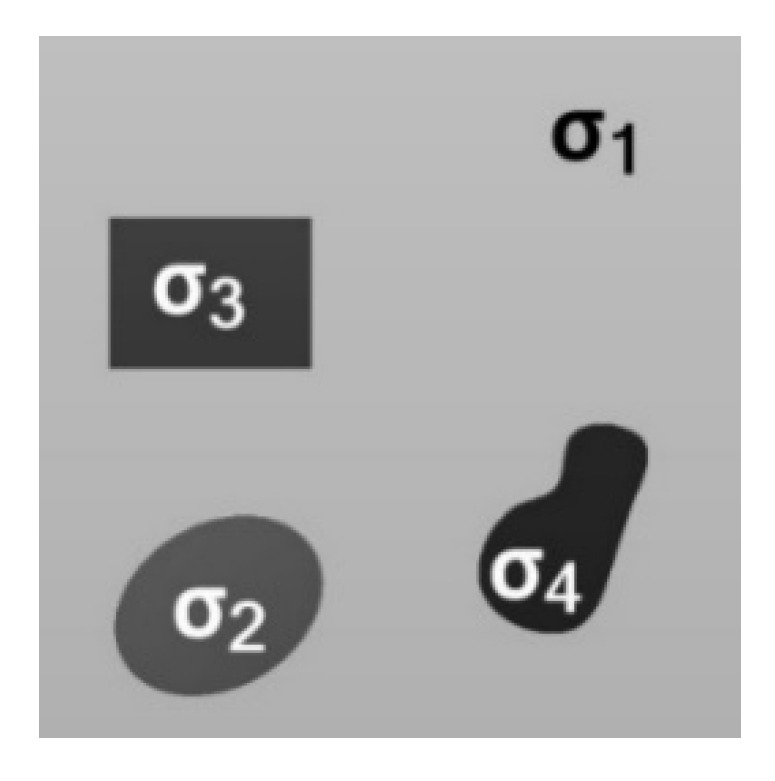
Multiple sub-pixel targets in a pixel; where σ1 is the roughness of the main region of the pixel, and σ2,3,4 denotes the roughness of the interfering sub-pixel regions.

**Figure 4 sensors-21-04418-f004:**
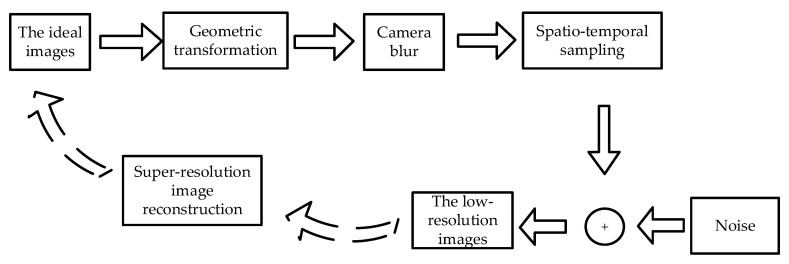
Model for generating low-resolution images.

**Figure 5 sensors-21-04418-f005:**
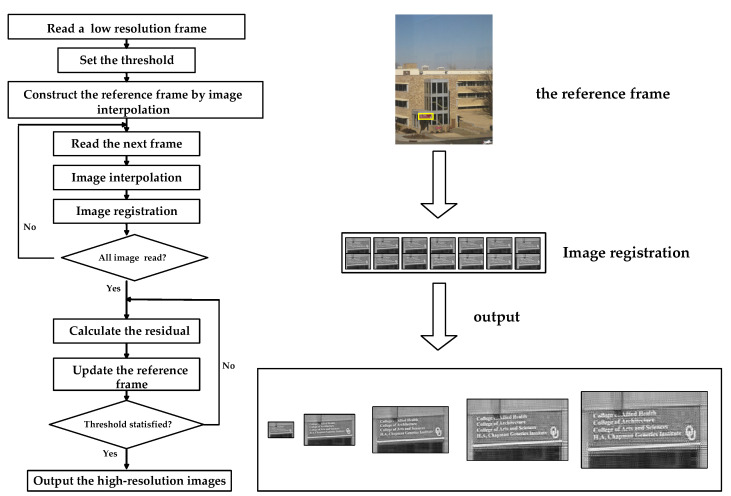
Super-resolution image reconstruction.

**Figure 6 sensors-21-04418-f006:**
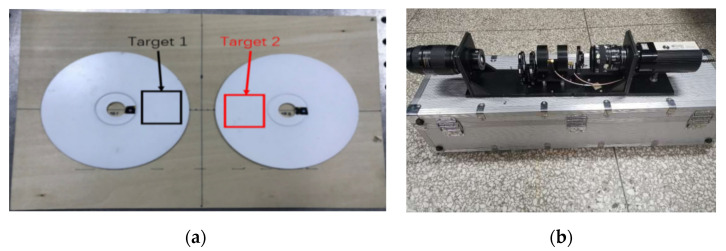
(**a**)Indoor experimental target. (**b**) LCVR-based polarized remote sensor.

**Figure 7 sensors-21-04418-f007:**
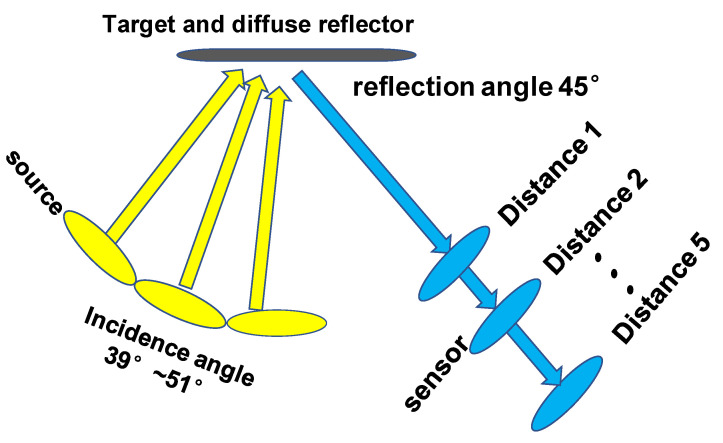
The experimental scheme.

**Figure 8 sensors-21-04418-f008:**
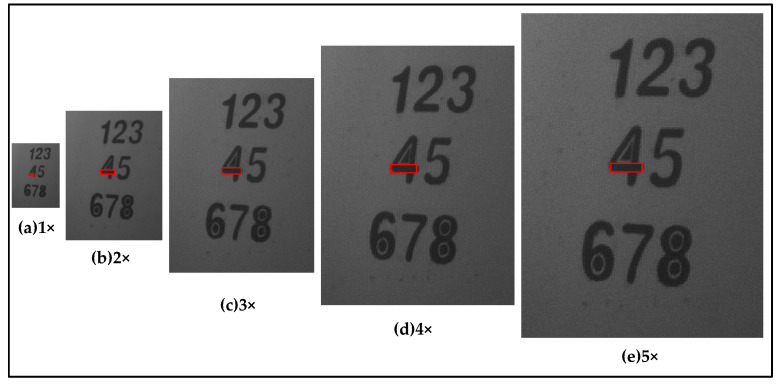
Images of target 3 at various magnifications. The inversion areas are highlighted by the red box.

**Figure 9 sensors-21-04418-f009:**
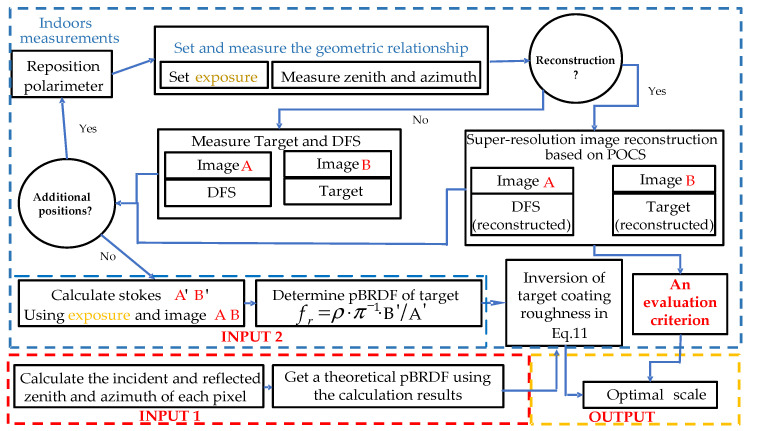
Scheme of experiment 2.

**Figure 10 sensors-21-04418-f010:**
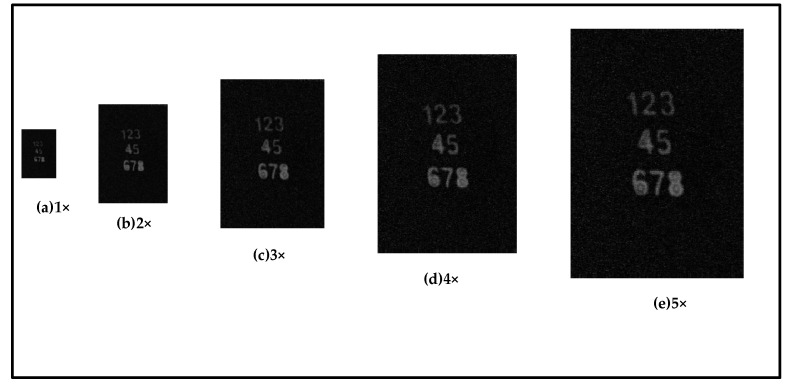
Reconstructed DOP images at various magnifications.

**Figure 11 sensors-21-04418-f011:**
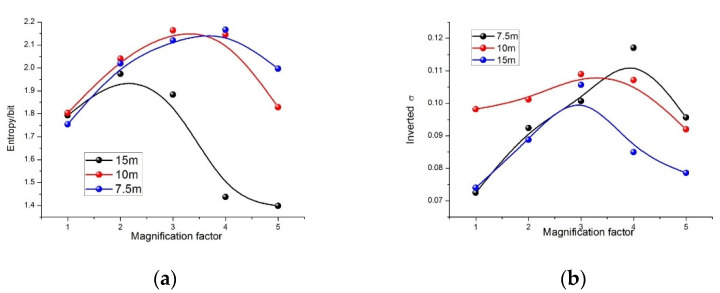
(**a**) Relation between entropy and magnification factor of the reconstructed DOP images. (**b**) Relation between the inverted σ and magnification factor of the reconstructed polarization images.

**Table 1 sensors-21-04418-t001:** Comparison, before and after DOP correction, of target 1 at an incident angle of 45°.

Group	Distance (cm)	The mean of DOP Original→Corrected → Change	The variance of DOP Original→Corrected → Change
1	596	0.6401→0.6390→0.17%	4.3907 × 10^−4^→3.4256 × 10^−4^→21.98%
2	326	0.6325→0.6315→0.16%	1.7482 × 10^−4^→1.5033 × 10^−4^→14.01%
3	236	0.6444→0.6437→0.11%	3.0338 × 10^−4^→2.8040 × 10^−4^→7.57%
4	174	0.6352→0.6350→0.03%	1.7069 × 10^−4^→1.5823 × 10^−4^→7.30%
5	115	0.6286→0.6280→0.09%	2.9821 × 10^−4^→2.6745 × 10^−4^→10.31%

**Table 2 sensors-21-04418-t002:** Roughness inversion results of laboratory experiments.

**Target 1**					
Group	Distance (cm)	Real σ	*σ_r_*	*σ_a_*	Relative error (%)
1	596	0.0543	0.0553	0.0551	1.763→1.530
2	326		0.0562	0.0558	3.438→2.796
3	236		0.0569	0.0568	4.864→4.621
4	174		0.0581	0.0581	7.064→6.926
5	115		0.0594	0.0570	9.380→4.976
**Target 2**					
Group	Distance(cm)	Real σ	*σ_r_*	*σ_a_*	Relative error (%)
1	596	0.0430	0.0454	0.0450	5.535→4.554
2	326		0.0449	0.0443	4.463→3.040
3	236		0.0445	0.0445	3.484→3.480
4	174		0.0451	0.0451	4.894→4.829
5	115		0.0463	0.0455	7.707→5.819

**Table 3 sensors-21-04418-t003:** Inverted σ of target 3.

Distance (m)	Magnification Factor	Measured σ	Inverted σ	Error
7.5	1	0.110	0.07248	34.11%
	2		0.09241	15.99%
	3		0.10071	8.45%
	4		0.11709	6.45%
	5		0.09562	13.07%
10	1		0.09817	10.75%
	2		0.1012	8.00%
	3		0.10895	0.95%
	4		0.10712	2.62%
	5		0.09199	16.37%
15	1		0.07401	32.72%
	2		0.08875	19.32%
	3		0.10565	3.95%
	4		0.08497	22.75%
	5		0.07856	28.58%

## Data Availability

The data presented in this study are available in article.

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
