# Peer review of "Spatial Scale Effect of a Typical Polarized Remote Sensor on Detecting Ground Objects"

_sensors, 2021, doi:10.3390/s21134418_

Round 1

Reviewer 1 Report

This manuscript tried to propose a method to reduce the spatial scale effects of the typical polarized remote sensor used for  featured and non-featured taegets.  The tested the effectiveness of considering the sub-pixel distribution analysis and the chosen of optimal scale with POCS method.  The comparison highlighted the importance of optical spatial scale chosen to reduce the sptial effects of polarized remote sensors. The proposed methods were accessible. It should be doubted, however,  that they used the linear-mixed pixel model to correct the spatial scale effects, since the complexity and non-linear correlations at the sub-pixel scale. 

Author Response

Response: Thanks for your comments and suggestions. In this manuscript, the spatial scale effect of the sensor on the roughness inversion of the coating was analysed from the perspective of spatial heterogeneity and our spatial scale effect correction method for linear mixed pixels was proposed. For the coatings lacking obvious features, the spatial scale effect caused by the difference in sub-pixel distribution was analysed. A scale correction formula based on roughness local variance is implemented to correct the inversion values obtained from the modified pBRDF model. However, for targets with obvious features, which are considered to be nonlinear mixed pixels, the correction formula is not suitable. Instead, we proposed a super-resolution reconstruction method to find the optimal scale through super-resolution reconstruction, based on the POCS method.

Reviewer 2 Report

The paper deals with a very important problem namely the effect of the spatial scale in the images obtained with a typical polarized remote sensor. The obtained images might differ due to the heterogeneity of objects and limited resolution. Two methods are proposed for two different types of object's coatings. Firstly, the scale correction formula based on the local variance of the roughness was implemented. Secondly, a new method based on POCS to find the optimal scale through super-resolution reconstruction was proposed. The experimental results showed that the proposed methods provide a way to reduce the spatial effect of the polarized remote sensor.

Thou the paper is very interesting and considers an important problem the presentation is to be enhanced before publication.

Major issues

The description of methods is to be enhanced.

line 167 eq6, 7 and 8 What does 'x' mean? Previously 'a dot' was used for multiplication. Further in eq9 '*' is used, probably for the same operation. A dot like in eq5 should be used for multiplication in all equations.

line 176 f with an arrow is called a matrix. What does the arrow mean for this matrix, other matrices don't have arrows?

line 195 eq11 is unclear. The nominator seems to be the sum of squared differences and the minus sign is given in the second row and there is a pair of square brackets. On the other hand there is another minus sign on the middle level on the right hand side of the first row.

line 264 - 274 In eq 21 C is introduced with 1 subscript for iteration step, but in eq 24 C has two subsripts. 
These subscripts are not described but taking into account other explanations they are the image coordinates.
So where how to add an iteration subscript to eq 24? How the eq 24 can be explained - on each iteration the closed convex set C includes the elements of the initial high-res image that satisfy what? What are M1 and M2? This is to be added to the method explanation. Also a note should be added on non-gaussian noise and unknown PSF.

line 274 eq 25 - no subscript l for g

line 384 Fig.9 There is no any label on the diagram that marks the start of the procedure. There are no incoming arrows for the bottom left block but the path from it finishes in bottom right block and doesn't go through the whole diagram.

The statistical significance is to be checked of results.

line 343 table 1 changes in means are to be tested for statistical significance

Conclusions are to be draw from the obtained results and the corresponding section added.

Minor issues

line 13 and 16 abbreviations pBRDF and POCS are to be described when first used

line 20 available method - reliable or viable

line 27 polarization can carry more information of objects - polarization can carry additional information about objects

line 36 With the widespread use and maturity of polarized remote sensing, polarization im-
aging in quantitative remote sensing has become one of the main trends in development,
regardless of polarization detection of near-surface or remote sensing platforms. - is hard to understand, please rephrase or split the sentence into pieces.

line 43 At multiple spatial scales - at different

line 116, 119, 123, 160, 196, 197, 205, 212, 237 [Error! Reference
source not found.]

line 118 A series of pBRDF models are based on the microfacet theory [Error! Reference source not found.]. - was already said in the previous phrase.

line 169 Combining the Mueller matrices of the instrument Mins - combining into

line 197 the upper index m is formatted for s though it should be an upper index for f

line 355 It is not the smaller the spatial resolution, the higher the inversion precision, but the highest inversion accuracy at the optimal resolution. - hard to understand, please rephrase

Reviewer 3 Report

Authors should take into account the following recommendations:

  • Acronyms should be avoided in the abstract
  • Acronyms must be defined the first time they appear in the text (only once).
  • When multiple examples or application cases are listed, a specific citation should be given for each example or case. If only one citation is indicated after enumeration, it should be valid for all the examples or applications listed. The first point where this problem appears is paragraph 1 of the article.
  • Authors should check all the text as multiple errors appear in the references to the number of equations (example: this happens 4 times on page 3).
  • Variables that appear in a figure but have not been defined in the text should be defined in the caption of said figure.
  • Authors should review the variables included in the text, as they do not exactly match the variables in the equations. In many cases the variable that appears in an equation with a subscript appears in the text without a subscript.
  • Equation (3) defines cos(2 beta), but equation (1) uses cos(beta).
  • Some variables in equation (11) do not match those listed in the text on lines 196 and 197.
  • The font used in equation (26) is very small, which makes its content difficult to read.
  • Authors should review the format of the References section, as it is not the same in all items.

Round 2

Reviewer 2 Report

The authors fixed all the issues except one - the conclusions are to be summarized in separate section named Conclusions.

Author Response

Response: Thanks for your comments and suggestions. The conclusions have been summarized in separate section named Conclusions in the manuscript.
